# The Utility of EQUAL Candida Score in Predicting Mortality in Patients with Candidemia

**DOI:** 10.3390/jof8030238

**Published:** 2022-02-27

**Authors:** Aline El Zakhem, Rozana El Eid, Rachid Istambouli, Hani Tamim, Souha S. Kanj

**Affiliations:** 1Division of Infectious Diseases, American University of Beirut Medical Center, Beirut 110236, Lebanon; az51@aub.edu.lb (A.E.Z.); rre10@mail.aub.edu (R.E.E.); 2Leeds and York Partnership NHS Foundation Trust, Leeds LS15 8ZB, UK; rachid_istambouli@hotmail.com; 3Biostatistics Unit, Clinical Research Institute, American University of Beirut, Beirut 110236, Lebanon; htamim@aub.edu.lb

**Keywords:** EQUAL score, candidemia, mortality, Candida

## Abstract

In an effort to standardize practice, the European Confederation of Medical Mycology (ECMM) developed the European Confederation of Medical Mycology Quality of Clinical Candidaemia Management (EQUAL) Candida score. This study investigated the utility of the EQUAL Candida score in predicting mortality in patients with candidemia admitted between January 2004 and July 2019. A total of 142 cases were included in the study, and 43.6% died within 30 days of candidemia diagnosis. There were no significant differences between survivors and non-survivors in terms of comorbidities predisposing to candidemia, except for malignancy (*p* = 0.021). The overall mean EQUAL score was 11.5 in the total population and 11.8 ± 3.82 and 11.03 ± 4.59 in survivors and non-survivors, respectively. When patients with a central venous catheter (CVC) were considered alone, survivors were found to have significantly higher scores than non-survivors (13.1 ± 3.19 vs. 11.3 ± 4.77, *p* = 0.025). When assessing components of the EQUAL Score separately, only candida speciation (*p* = 0.013), susceptibility testing (*p* = 0.012) and echocardiography results (*p* = 0.012) were significantly associated with a lower case-fatality rate. A higher EQUAL Candida score was able to predict a lower case-fatality rate in patients with a CVC.

## 1. Introduction

The incidence of candidemia has been rising worldwide over the last decades, amounting to be the fourth highest bloodstream infection in hospitalized patients [1]. This is believed to be due to the increase in predisposing factors, such as immunosuppression, critical illness, advanced age, the use of broad-spectrum antibiotics, as well as indwelling venous catheters [2,3,4,5]. Moreover, despite the evolution of treatment strategies, associated mortality remains significant, with 30-day mortality reaching up to 60% [2], often accompanied by a long hospital stay and a significant economic burden [1].

The management of candidemia and guidelines have been evolving over the past few years to address certain controversial areas [6,7,8]. Currently, empirical therapy with an echinocandin is strongly recommended, with high-quality evidence [6], with step-down therapy to an azole agent when possible, according to isolate speciation and antifungal susceptibility testing. The main remaining controversial points in candidemia management include the indications for ophthalmic examination and echocardiography as well as the need for removal of central lines in patients with central line-associated bloodstream infection [9].

The European Confederation of Medical Mycology (ECMM) developed the European Confederation of Medical Mycology Quality of Clinical Candidaemia Management (EQUAL) score to assess the adherence to the latest guidelines in both the diagnosis and the management of candidemia and to highlight the most important aspects of the current recommendations [10]. The components of the score were given points according to the impact on patients’ outcomes and the strength of evidence.

The aim of this study is to evaluate adherence to guidelines in the diagnosis and management of candidemia and to assess the utility of the EQUAL candida score in predicting mortality in patients with candidemia.

## 2. Research Design and Methods

### 2.1. Patient Identification

A single-center retrospective chart review was conducted on all patients with candidemia admitted from January 2004 until July 2019 at the American University of Beirut Medical Center (AUBMC), a tertiary care center in Lebanon. All episodes of candidemia were identified using Electronic Health Records (EHR). Only the first episode of candidemia was considered for a patient with recurrent candidemia. A positive blood culture for any *Candida* species isolated after a period of time greater than 30 days from the initial episode of candidemia was considered as a new episode. Inclusion criteria were patients with age ≥ 18 and having proven candidemia by isolation of *Candida* spp. in at least one blood culture. In order to estimate attributable mortality due to candidemia, patients who died before the results of the blood cultures or within 48 h of results were excluded from the analysis. In addition, patients with bacterial bloodstream co-infection were excluded from the study.

### 2.2. Data Extraction

Data were collected on patient characteristics, including age, sex, underlying co-morbid conditions, and known risk factors for candidemia (diabetes, presence of central venous catheter (CVC), critical illness, immunosuppression, hemodialysis, abdominal surgery, history of antibiotic therapy in the last 30 days, history of antifungal therapy in the last 30 days, malignancy), and on the source of candidemia. Immunocompromised patients included those with malignancy, immune deficiency disorders, and conditions that alter a patient’s normal immune function, such as burns, as well as those taking immunosuppressive therapy for rheumatologic diseases or inflammatory bowel diseases.

Data on 30-day mortality were also collected. Additionally, data on the components of the EQUAL Candida score were collected. Score points were allocated as follows: initial blood cultures (40 mL) (3 points), species identification (3 points), susceptibility testing (2 points), echocardiography performed (1 point), ophthalmoscopy performed (1 point), follow-up blood culture (at least one per day until negative) (2 points) and treatment for 14 days after first negative follow-up culture (2 points). If the patient died while on antifungal therapy within the first 14 days, 2 points were allocated. Additionally, initial echinocandin treatment was allocated 3 points, and step-down therapy to fluconazole depending on susceptibility result received 2 points. If fluconazole was the initial antifungal agent used, 0 points were allocated. If the *Candida* isolate was resistant to fluconazole, 2 points were allocated if echinocandin was continued or if there was step-down therapy to voriconazole. In patients with CVC at the onset of candidemia, CVC removal ≤24 h from diagnosis received 3 points, while those removed >24 but ≤72 h from diagnosis was allocated 2 points. The maximal score for patients with a CVC is 22 and those without CVC is 19 (Table 1).

### 2.3. Statistical Analysis

Statistical analysis was performed using IBM SPSS (IBM Corp., New York, NY, USA). Categorical data is displayed as frequencies and percentages, and continuous data is presented as means and standard deviation for normally distributed variables. Univariate analysis for comparisons between survivors and non-survivors was calculated using the student *t*-test for continuous variables and the Chi-square test for non-continuous variables. Variables with a *p*-value < 0.05 on comparison analysis were included in a stepwise multiple logistic regression analysis to determine risk factors associated with 30-day mortality. A *p*-value < 0.05 was considered statistically significant. Model performance was evaluated with receiver operating characteristic (ROC) analysis to determine a cut-off score. The area under the curve (AUC) was used as a single performance measure to decide whether the model prediction was better than random (0.5). A perfect model would yield an AUC value of 1.

### 2.4. Ethical Considerations

The study has received approval by the institutional review board (IRB) at AUBMC (Protocol number: BIO-2019-0290). Patient consent was waived, as this is a retrospective chart review. The study included all adult patients with candidemia presenting to AUBMC in the study period, with no regard to sex and ethnic background. It posed no more than minimal risk to patients. The potential benefits of the study outweigh the potential risks.

## 3. Results

A total of 142 cases with candidemia were identified between January 2004 and July 2019, representing 138 patients [5]. The majority were immunocompromised (86%) and less than half of them (44%) were admitted to a critical care unit at the time of diagnosis. Most of the patients had a CVC (71%) at the time of diagnosis. Table 2 shows a summary of baseline patients’ characteristics.

In total, 62 patients (43.6%) died within 30 days of candidemia diagnosis. There was no significant difference in the median age between survivors and non-survivors (62 ± 19 vs. 68 ± 15, *p* = 0.061). There were no significant differences between survivors and non-survivors in terms of comorbidities predisposing to candidemia, except for malignancy (*p* = 0.021) (Table 2). At the time of diagnosis of candidemia, non-survivors were more likely to be critically ill (58.1% vs. 32.5% in survivors, *p* = 0.002). *Candida albicans* was the most frequently detected *Candida* isolate, found in 51/142 cases, three of which had a mixed Candida infection with *Candida glabrata*, *Candida dubliniensis*, and *Candida tropicalis*, respectively. However, as a group, non-albicans *Candida* (NAC) predominated, with the most commonly detected isolate being *C. glabrata* (26.1%). However, there was no significant difference in the case-fatality rate between patients infected with *albicans* vs. non-*albicans* isolates. Even though gastrointestinal translocation was the most common identifiable source of candidemia, it was not associated with significant differences between survivors and non-survivors, as opposed to unknown etiology of the candidemia and urinary tract infection (UTI) as a source (*p* = 0.002 and *p* = 0.031, respectively).

Antifungal susceptibility testing was performed after the diagnosis of candidemia in more than half of the patients (57%). While echocardiography was performed in 57% of patients, ophthalmoscopy was done in only 23.9% of patients. Data were available for 130 out of the 142 cases regarding initial antifungal therapy, whereby empiric echinocandin treatment was given to around half of the patients (50.8%). Appropriate step-down therapy to fluconazole after identifying fluconazole-susceptible isolates was carried out in approximately 20% of patients after initial echinocandin treatment. Treatment for 14 days after the first negative follow-up blood culture was completed in 72 patients (50.7%). Only 24 of the patients had daily follow-up blood cultures. CVCs were present in 101 of 142 cases and was removed in the majority of them (77%). These frequencies are depicted in Figure 1.

The overall mean EQUAL score was 11.5 in the total population and was not statistically different between survivors and non-survivors. However, in the subset of patients with CVC, survivors were found to have significantly higher scores than non-survivors (13.1 ± 3.19 vs. 11.3 ± 4.77, *p*= 0.025). When assessing components of EQUAL score separately, only speciation (*p* = 0.013), susceptibility testing (*p* = 0.012) and echocardiography (*p* = 0.012) were associated with lower case-fatality rate. A summary of the different variables stratified by case-fatality rate is shown in Table 3. 

In the multivariable logistic regression analysis, mechanical ventilation (odds ratio (OR) 3.141, *p* = 0.002), immunocompromised status (OR 2.666, *p* = 0.016), and unknown source of candidemia (OR 2.269, *p* =0.048) were significant predictors of mortality when considering the total patient population. Factors such as malignancy which were significant on bivariate analysis were not significant as per the regression analysis and were statistically excluded from the model. Additionally, in the subset of patients with CVC, mechanical ventilation (OR 3.360, *p*= 0.008) and unknown source of candidemia (OR 4.270, *p* = 0.006) contributed to the model significantly, whereas the EQUAL score showed only borderline significance in the final model (OR 0.892, *p* = 0.050) (Table 4).

The ROC curve revealed an AUC of 0.55 for the regression model including the total patient population. Though this shows very poor prediction ability, a cutoff point of 7.5 was found to be significant (Figure 2).

## 4. Discussion

Since the introduction of the proposed ECMM EQUAL score, only a limited number of studies have been published investigating its internal applicability and usefulness in clinical practice [10,11,12,13,14,15]. Moreover, this is the first study to address the validity of the score in patients with candidemia from the eastern Mediterranean region. Our results revealed that practicing physicians at our institution had varying levels of adherence to candidemia guidelines measured through the EQUAL score, ranging from 100% for initial blood cultures and 83% for species identification to 16.9% for daily follow-up blood cultures until negative. Among the total patient population, the overall mean score was 11.5. The mean EQUAL score found in this study is higher than others found in the literature, ranging from a mean score of 8.91 from a multicenter study in Taiwan [13] to 9.9 and 11.04 from studies done in Germany and the United Kingdom, respectively [14,16]. A median EQUAL score of 6 (Inter-quartile Range (IQR) 6-9) for patients without CVCs and 11 (IQR 6-14) for patients with CVCs was reported in a Portuguese tertiary care hospital [12], whereas, in Korea, a median EQUAL score of 17 (14–18) for patients with CVCs and 14 (12–15) for patients without CVCs were reported [15]. These fluctuating variations globally may attest to the disputed nature of the management of candidemia in terms of a cost-effective strategy or in the setting of limited healthcare resources, particularly in low- and middle-income countries. Various studies show a wide difference in the adherence to the current guidelines in the countries studied. While species identification was uniform at 100%, antifungal susceptibility testing, for example, ranged from 1.7% in Portugal to 100% in Germany, and ophthalmic examinations suffered from poor adherence with only 1.9–30.5% [12,14]. This further emphasizes the need for a unifying tool that encourages physicians to follow best practice guidelines.

In many of these studies, the EQUAL score was assessed for its impact on mortality, and this has yielded conflicting results. Our results revealed a significant difference in patients with a CVC in particular, where survivors had significantly higher scores than non-survivors (13.1 ± 3.19 vs. 11.3 ± 4.77, *p* = 0.025). Similar to our findings, Huang et al. [13] found a significant difference between the EQUAL score in survivors and non-survivors in the subgroup of patients with CVCs, and the scores did not differ significantly in patients without CVCs. A study from Portugal [12] showed no statistically significant association between EQUAL Candida Score and mortality. They additionally found a significant association in the overall population (combining patients with a CVC and those without). Conversely, the results from a Korean tertiary care center showed no difference in mortality across all groups except when applying a cut off score of <15 for patients overall or the subgroup with a CVC. Huang et al. [13] reported that those with EQUAL scores of more than 10 had a significantly higher survival rate than those with scores <10. These differing cut-offs of the score and variable results are difficult to interpret clinically, and more extensive studies are needed to better assess the utility of the EQUAL score in optimizing patient outcomes.

When looking at the components of the score individually, we found echocardiography to be a significant predictor of mortality. Though many guidelines strongly advise it [17], in reality, very few studies assessed the yield of systematically doing echocardiography to rule out endocarditis in all patients with candidemia. One large prospective study revealed a 4.2% prevalence of candida endocarditis in patients with candidemia [18]. As for the ophthalmologic exam, our results showed that it was not correlated with lower mortality. The IDSA and many other expert bodies advocate for routine ophthalmoscopy in all patients [6]. However, this recommendation has been revised based on some studies [19] indicating a low frequency of ocular involvement and favorable outcomes without it, given the recent advances in treatment. Our study showed that out of the 142 cases, 34 underwent ophthalmologic evaluation, only three patients were found to have evidence of endophthalmitis. This is similar to other studies where endophthalmitis was found in a low percentage of cases, and performance of fundoscopy was not associated with improved survival [11]. However, further data is needed to establish the highest risk population that would most benefit from this intervention.

The susceptibility pattern of our *Candida* isolates was much more favorable than has been recently reported in the literature. Of the overall isolates, 75% were susceptible to fluconazole and 89% susceptible to voriconazole, while 100% of the isolates were susceptible to echinocandins. Compared to one surveillance study done in Denmark, fluconazole susceptibility was shown to be decreasing (68.5%, 65.2%, and 60.6% in 2004 to 2007, 2008 to 2011, and 2012 to 2015, respectively, *p* < 0.0001) [20]. Additionally, there was an increasing trend of fluconazole resistance among isolated *Candida* spp. in a study from Korea, particularly for *Candida parapsilosis* isolates, a finding commonly reported in intensive care units [15,21,22]. *C. parapsilosis* has also demonstrated recent resistance to echinocandins [11]. The non-significant association between case-fatality rate and detection of albicans versus non-albicans isolates in our study could be due to the low proportion of *C. parapsilosis*. In addition, the data was collected prior to the emergence of the *Candida auris* outbreak at our medical center [23], which could have significantly affected the mortality rates when comparing *albicans* to non-*albicans* given the expected high mortality of *C. auris*.

To prevent further emergence of antifungal resistance, experts in the field have advocated for directed antifungal therapy based on routine susceptibility testing and step-down therapy based on those results. In our cohort, less than 20% of patients were switched to fluconazole after susceptibility results revealed that the isolate was susceptible. Infectious diseases consultations have been shown in a recent systematic review and meta-analysis to reduce mortality of candidemia [24,25]. This is highly recommended, particularly in critically ill patients, where applying antifungal stewardship principles, including de-escalation, are much needed [26].

Regarding the variables included in the final regression model of our study, it is unsurprising that surrogate markers of clinical severity such as mechanical ventilation were found to be highly predictive of mortality. Also, the source of candidemia can affect mortality. Similar to our findings, Keighley et al. developed a predictive model stratifying patients with candidemia into <20% and ≥20% 30-day mortality and found that a gastrointestinal or unknown source of candidemia was associated with higher overall mortality than an intravascular or urologic source [27]. Increased mortality due to an unknown source of candidemia may be due to delayed diagnosis. Many of these patients may not have the conventional risk factors predisposing to candidemia, stressing the greater need to have a low threshold of suspicion, especially in critically ill patients.

Our study has limitations due to its retrospective nature and relatively low patient population. Additionally, during the 15 years, there might have been minor differences in management practices between attending physicians that may have affected outcomes. Nevertheless, the study represents an important addition to the literature as it reinforces that adherence to the guidelines set through the EQUAL score is correlated with a lower case-fatality rate in patients with a CVC. This subset of patients is more likely to be critically ill, and thus efficient and quick management is of paramount importance. Therefore, clinicians may use the EQUAL score not only as a checklist to ensure the best quality care is being delivered but also as a prediction tool to infer prognosis. There is a need for this score to be validated by prospective or randomized studies, especially in patients without a CVC.

In conclusion, this study evaluated current practices in the management of candidemia using the EQUAL score as a tool, highlighting various gaps, and suggesting improved strategies. Though the EQUAL score did not show any significant association with mortality in the overall candidemia group, it may be a useful framework when considering patients with a CVC. Optimal ophthalmic examination strategies and antifungal treatment in different clinical situations should be assessed in future studies.

## Figures and Tables

**Figure 1 jof-08-00238-f001:**
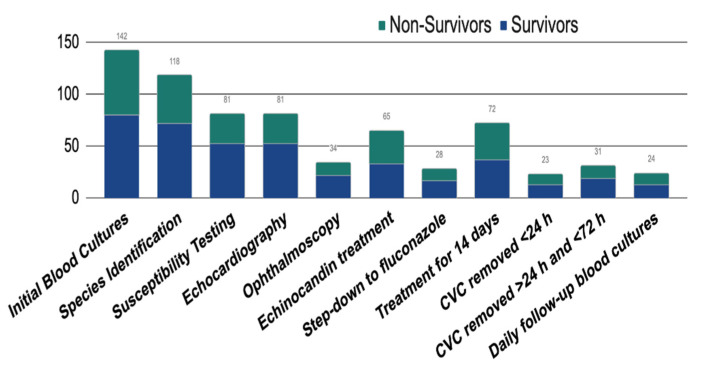
Adherence to quality indicators in 142 cases. Abbreviations: CVC: central venous catheter, h: hours.

**Figure 2 jof-08-00238-f002:**
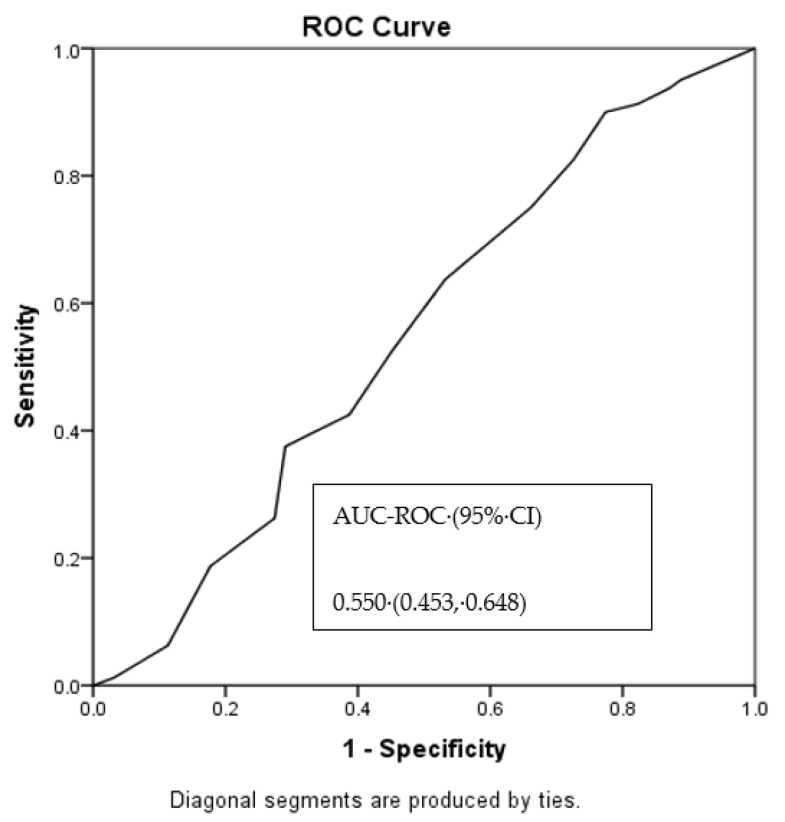
AUC-ROC Curve Analysis for the multivariable logistic regression model, including the total population. Abbreviations: CI, confidence interval; AUC-ROC, area under the curve-receiver operating characteristic.

**Table 1 jof-08-00238-t001:** EQUAL Candida Score ^1^.

	Score ^1^	
Quality Indicator	Patients with CVC ^2^	Patients without CVC ^2^
Initial blood culture (40mL)	3	3
Species identification	3	3
Susceptibility testing	2	2
Echocardiography	1	1
Ophthalmoscopy	1	1
Echinocandin treatment	3	3
Step down to fluconazole depending on susceptibility result	2	2
Treatment for 14 days after first negative follow-up culture	2	2
CVC removal		n/a
≤24 h from diagnosis	3	
24–72 h from diagnosis	2	
Follow-up blood culture (at least one per day until negative)	2	2
Maximum score	22	19

^1^ Mellinghoff SC, Hoenigl M, Koehler P, Kumar A, Lagrou K, Lass-Florl C, et al. EQUAL Candida Score: An ECMM score derived from current guidelines to measure QUAlity of Clinical Candidaemia Management. *Mycoses*
**2018**, *61*, 326–330. ^2^ Central Venous Catheter.

**Table 2 jof-08-00238-t002:** Baseline characteristics of the enrolled patients.

Characteristics	n (%)
Age (mean ± SD)	64.8 ± 17.5
Sex, Female	68 (47.9%)
Abdominal Surgery in last 30 days	38 (26.8%)
Antibiotic Use ^&^	137 (96.5%)
Antifungal Use ^&^	39 (27.5%)
Immunocompromised	93 (65.5%)
Solid Transplant	2 (1.4%)
Malignancy	76 (53.5%)
HSCT ^1^	6 (4.2%)
Diabetes	41 (28.9%)
Hemodialysis	21 (14.8%)
Clinical Severity	
Neutropenia	14 (9.9%)
Critical Care Admission	62 (43.7%)
Parenteral Nutrition	23 (28.7%)
*Candida* Species	
non-*albicans Candida*	94 (66.2%)
*C. albicans*	48 (33.8%)
Central venous catheter	101 (71.1%)
CVC ^2^ removal	78 (54.9%)
CVC ^2^ retention	23 (16.2%)
Source of candidemia	
CLABSI ^3^	36 (25.4%)
UTI ^4^	13 (9.2%)
GI ^5^	49 (34.5%)
Unknown	43 (30.3%)
EQUAL Candida Score (mean± SD)	11.5 ± 4.18

Abbreviations: ^1^ HSCT: hematopoietic stem cell transplantation, ^2^ CVC: Central venous catheter, ^3^ CLABSI: central line-associated bloodstream infection, ^4^ UTI: urinary tract infection, ^5^ GI: gastrointestinal, **^&^** defined as intake within 30 days from candidemia episode.

**Table 3 jof-08-00238-t003:** Comparative characteristics of the 142 cases of candidemia stratified by case-fatality rate.

30-Day Case-Fatality Rate			
	Survivors *N* = 80, n/N (%)	Non-Survivors *N* = 62, n/N (%)	*p*-Value
Age (mean ± SD)	62 ± 19	68 ± 15	0.061
Sex, Female	40 (50.0%)	28 (45.2%)	0.567
Abdominal Surgery	23 (28.7%)	15 (24.2%)	0.543
History of antibiotics intake ^&^	77 (96.3%)	60 (96.8%)	1.000
History of antifungal intake ^&^	18 (22.5%)	21 (33.9%)	0.132
Comorbidities			
Diabetes	21 (26.3%)	20 (32.3%)	0.433
Hemodialysis	12 (15.0%)	9 (14.5%)	0.936
Solid Transplant	0 (0.0%)	2 (3.2%)	0.189
Malignancy	36 (45.0%)	40 (64.5%)	0.021 *
HSCT ^1^	3 (3.8%)	3 (4.8%)	1.000
Immunocompromised	44 (55%)	79 (42.9%)	0.003
Clinical Severity			
Neutropenia	5 (6.3%)	9 (14.5%)	0.101
Critical care admission	26 (32.5%)	36 (58.1%)	0.002 *
Parenteral Nutrition	23 (28.7%)	17 (27.4%)	0.861
*Candida* Species			0.076
non-*albicans Candida*	48 (60.0%)	46 (74.2%)	
*C. albicans*	32 (40.0%)	16 (25.8%)	
Central venous catheter (*N* = 101)			<0.001 *
CVC ^2^ removal	49 (92.5%)	29 (60.4%)	
CVC ^2^ retention	4 (7.5%)	19 (39.6%)	
Source of candidemia			
CLABSI ^3^	24 (30.0%)	12 (19.4%)	0.148
UTI ^4^	11 (13.8%)	2 (3.2%)	0.031 *
GI ^5^	28 (35.0%)	21 (33.9%)	0.888
Unknown	16 (20.0%)	27 (43.5%)	0.002 *
EQUAL Candida Score (mean± SD)	11.8 ± 3.82	11.03 ± 4.59	0.256

* Denotes significant *p*-values of <0.05. Abbreviations: ^1^ HSCT: haematopoietic stem cell transplantation, ^2^ CVC: Central venous catheter, ^3^ CLABSI: central line-associated bloodstream infection, ^4^ UTI: urinary tract infection, ^5^ GI: gastrointestinal, ^&^ defined as intake within 30 days from candidemia episode.

**Table 4 jof-08-00238-t004:** Multivariable logistic regression of factors associated with the 30-day case-fatality rate after diagnosis of candidemia.

	OR	95% C.I.	*p*-Value
**Total Population**			
EQUAL score	0.949	0.867–1.038	0.250
Mechanically ventilated	3.141	1.497–6.588	0.002
Immunocompromised	2.666	1.197–5.936	0.016
Unknown source of candidemia	2.269	1.009–5.104	0.048
**CVC Carriers**			
EQUAL score	0.892	0.796–1.000	0.050
Mechanically ventilated	3.360	1.370–8.242	0.008
Unknown source of candidemia	4.270	1.515–12.034	0.006

Abbreviations: CVC: central venous catheter, OR: odds ratio, CI: Confidence Interval.

## Data Availability

The data presented in this study are available on request from the corresponding author. The data are not publicly available to ensure privacy of data collected from patients.

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
