# Peer review of "The Utility of EQUAL Candida Score in Predicting Mortality in Patients with Candidemia"

_jof, 2022, doi:10.3390/jof8030238_

Round 1

Reviewer 1 Report

Urinary tract infection was found to be the source of Candidemia in 9.2% of your patients. Please detail how many patients had Candida pyelonephritis and how many had only lower urinary infection, caused by Candida spp?

Do you think that the routine evaluation for patients with Candidemia in your hospital, were the same from 2004 to 2019? If not, this could be impact or interfere in your 15 years analysis?

Author Response

  1. Urinary tract infection was found to be the source of Candidemia in 9.2% of your patients. Please detail how many patients had Candida pyelonephritis and how many had only lower urinary infection, caused by Candida spp?

Given the low number of patients (13) we did not make the distinction between upper and lower UTI as this would not have been statistically significant to interpret.

  1. Do you think that the routine evaluation for patients with Candidemia in your hospital, were the same from 2004 to 2019? If not, this could be impact or interfere in your 15 years analysis?

Yes, since all patients enrolled during this time were under the same infectious disease leadership and stewardship program.

Reviewer 2 Report

The author reported the association of EQUAL score with the case-fatality rate in candidemia patients. However, as written, the result does not match the aim of the study but rather focuses on the general description of the candidemia patients.

  1. The author should focus the result more on the association between EQUAL score and the case-fatality rate.
  2. The sample size calculation is needed for the prediction study.
  3. The association between EQUAL score and case-fatality rate is not significant (0.256). Therefore, the followed regression analysis and ROC curve should not be performed.
  4. Some significant factors such as UTI and malignancy in table 3 were not presented in table 4. If they are not significant in the subsequent analysis, please mentioned this briefly in the text.
  5. Case fatality rate should be used instead of mortality rate.

Author Response

  1. The author should focus the result more on the association between EQUAL score and the case-fatality rate.

The association between EQUAL score and mortality was mentioned extensively (lines 150-155) in addition to the tables which report on the statistical results, both for the EQUAL score and for its individual components. In addition, it is important to report all factors that may additionally contribute to death so as not to misguide clinical interpretation.

  1. The sample size calculation is needed for the prediction study.

Since this was a retrospective study, we do not have a pre-determined sample size.

  1. The association between EQUAL score and case-fatality rate is not significant (0.256). Therefore, the followed regression analysis and ROC curve should not be performed.

We decided to do the multivariate analysis for the association between the EQUAL score and mortality to exclude a scenario of a confounding effect by the other factors which could have caused this insignificant association.

  1. Some significant factors such as UTI and malignancy in table 3 were not presented in table 4. If they are not significant in the subsequent analysis, please mentioned this briefly in the text.

Some factors such as malignancy were significant but when included in the in the multivariable logistic regression analysis, they turned out to be not significant and thus were excluded from the final regression model. This was reinforced in lines (168-170).

  1. Case fatality rate should be used instead of mortality rate.

This has been changed in the text in the appropriate contexts.

Reviewer 3 Report

The author aimed to investigate whether EQUAL candida score can be used to predict mortality or not. Although the author did a good job in conducting this study, there are many issues to be discussed.

  1. I think the most important drawback of this study is the study design. The author aimed to investigated the utility of the EQUAL candida score in predicting mortality among patients with candidemia. However, from the study design and the results, the author is analyzing the clinical characteristics and treatment outcomes of candidemia among patients from a single center in Lebanon.
  2. The author did find the factors associated with the 30-day mortality after diagnosis of candidemia using multivariable logistic regression (Table 4), and EQUAL score was not the independent factor. Therefore, the author can only conclude that in the presence of a CVC, the EQUAL score can be used. This result will not contribute to the literature. The author should conclude that EQUAL score will not be practical to predict mortality in patients with candidemia, since nearly 30% of patients had no CVCs.
  3. Why did the author make an AUC-ROC curve for multivariable logistic regression model in Figure 2? I think it is nonsense. The author did not mention the figure 2 in the text, and only the footnote of Table 4 mentioned the figure 2. This is inappropriate. Additionally, if the AUC is only 0.55, it is nonsense to find the cutoff point.
  4. The English of this manuscript is very poor. For example: “The incidence of candidemia has witnessed……” This is an inappropriate sentence, because how can the incidence witness? “This is believed to be due to”(line 28); “associated mortality…” and “Elements of the score were given points….” are all inappropriate writing.
  5. Some statements are very strange. Line 34: “management of candidemia remains controversial….” Why management of candidemia remains controversial? I do not agree current guideline recommend using echinocandin as empirical therapy. (line 35).
  6. The author applied a single-center study to investigate the application of EQUAL candida score, with data from long study period “2004-2019”. During the 15 years, there have been differences in the therapeutic strategies and new antifungal agents. Maybe the authors needed a large number of cases to complete the study. However, there will be bias for such study because so many things will be changed during this long period. At least this issue should be mentioned in the limitation of this study.
  7. The case number of this study is very small. For a study aim of verifying an important score, a much more case number is needed to complete this work.
  8. Why does the patient with bacterial bloodstream co-infection were excluded? I think it is quite common (although maybe it is less than 20%) to have combined bacteremia/candidemia in clinical practice.

Author Response

  1. I think the most important drawback of this study is the study design. The author aimed to investigated the utility of the EQUAL candida score in predicting mortality among patients with candidemia. However, from the study design and the results, the author is analyzing the clinical characteristics and treatment outcomes of candidemia among patients from a single center in Lebanon.

The association between EQUAL score and mortality was mentioned extensively (lines 150-155) in addition to the tables which report on the statistical results, both for the EQUAL score and for its individual components. In addition, it is important to report all factors that may additionally contribute to death so as not to misguide clinical interpretation.

  1. The author did find the factors associated with the 30-day mortality after diagnosis of candidemia using multivariable logistic regression (Table 4), and EQUAL score was not the independent factor. Therefore, the author can only conclude that in the presence of a CVC, the EQUAL score can be used. This result will not contribute to the literature. The author should conclude that EQUAL score will not be practical to predict mortality in patients with candidemia, since nearly 30% of patients had no CVCs.

We agree with this statement, and we have reformulated our concluding statement to better clarify this (lines 286-288).

  1. Why did the author make an AUC-ROC curve for multivariable logistic regression model in Figure 2? I think it is nonsense. The author did not mention the figure 2 in the text, and only the footnote of Table 4 mentioned the figure 2. This is inappropriate. Additionally, if the AUC is only 0.55, it is nonsense to find the cutoff point.

Although it was not significant, we considered the equal score as a continuous variable and thus by doing the AUC we wanted to check if there is any cut off point that could identify a higher or lower risk of mortality. Also, the figure was mentioned in text however during processing of the article the font was reduced and appeared to be part of the footnote. This was corrected in the manuscript.

  1. The English of this manuscript is very poor. For example: “The incidence of candidemia has witnessed……” This is an inappropriate sentence, because how can the incidence witness? “This is believed to be due to”(line 28); “associated mortality…” and “Elements of the score were given points….” are all inappropriate writing.

We have revised the English language of the manuscript and made appropriate changes, for example in line 48 and line 231.

  1. Some statements are very strange. Line 34: “management of candidemia remains controversial….” Why management of candidemia remains controversial? I do not agree current guideline recommend using echinocandin as empirical therapy. (line 35).

Management guidelines are controversial in specific issues such as performance of echocardiography, which we further clarified in the text (lines 34-35). Echinocandin as empirical therapy has been recommended by the latest guidelines.

  1. The author applied a single-center study to investigate the application of EQUAL candida score, with data from long study period “2004-2019”. During the 15 years, there have been differences in the therapeutic strategies and new antifungal agents. Maybe the authors needed a large number of cases to complete the study. However, there will be bias for such study because so many things will be changed during this long period. At least this issue should be mentioned in the limitation of this study.

This is a valid point, added to the limitations (lines 276-277). To note that all patients enrolled during this time were under the same infectious disease leadership and stewardship program, and therefore we do not expect that this has affected the results significantly.

  1. The case number of this study is very small. For a study aim of verifying an important score, a much more case number is needed to complete this work.

While our sample size is limited, since our country is small with a low population density, this result can be applied to other countries especially in the Middle East that also have a similar demographic. In addition, we do not have a pre-determined sample size since this is a retrospective study. 

  1. Why does the patient with bacterial bloodstream co-infection were excluded? I think it is quite common (although maybe it is less than 20%) to have combined bacteremia/candidemia in clinical practice.

So as not to conflate mortality due to candidemia with mortality due to bacteremia/sepsis.

Round 2

Reviewer 2 Report

I am satisfied with all the responses.

However, the sample size calculation must be performed please see https://pubmed.ncbi.nlm.nih.gov/35138673/. 

Should the sample size is not sufficient, the author can write as a limitation in the discussion.